# Inhibition of Heme Oxygenase Antioxidant Activity Exacerbates Hepatic Steatosis and Fibrosis *In Vitro*

**DOI:** 10.3390/antiox8080277

**Published:** 2019-08-05

**Authors:** Marco Raffaele, Giuseppe Carota, Giuseppe Sferrazzo, Maria Licari, Ignazio Barbagallo, Valeria Sorrenti, Salvatore S. Signorelli, Luca Vanella

**Affiliations:** 1Department of Drug Science, Biochemistry Section, University of Catania, 95125 Catania, Italy; 2Department of Clinical and Experimental Medicine, University of Catania, 95125 Catania, Italy

**Keywords:** heme oxygenase, HO-1 inhibitor, NAFLD, hepatocytes, collagen, oxidative stress

## Abstract

The progression of non-alcoholic fatty liver disease (NAFLD) and the development of hepatic fibrosis is caused by changes in redox balance, leading to an increase of reactive oxygen species (ROS) levels. NAFLD patients are at risk of progressing to non-alcoholic steatohepatitis (NASH), associated to cardiovascular diseases (CVD), coronary heart disease and stroke. Heme Oxygenase-1 (HO-1) is a potent endogenous antioxidant gene that plays a key role in decreasing oxidative stress. The present work was directed to determine whether use of an inhibitor of HO-1 activity affects lipid metabolism and fibrosis process in hepatic cells. Oil Red assay and mRNA analysis were used to evaluate the triglycerides content and the lipid metabolism pathway in HepG2 cells. ROS measurement, RT-PCR and Soluble collagen assay were used to assess the intracellular oxidant, the fibrosis pathway and the soluble collagen in LX2 cells. The activity of HO-1 was inhibited using Tin Mesoporphyrin IX (SnMP). Our study demonstrates that a non-functional HO system results in an increased lipid storage and collagen release in hepatocytes. Consequently, an increase of HO-1 levels may provide a therapeutic approach to address the metabolic alterations associated with NAFLD and its progression to NASH.

## 1. Introduction

Hepatic steatosis is a common liver disease characterized by the presence of triglycerides vesicles, accumulating within hepatocytes [1]. It is associated with dyslipidaemia, obesity and insulin resistance, despite a diet with low alcoholic drink consumption: this condition is known as non-alcoholic fatty liver disease, or NAFLD [2,3,4,5,6,7]. NAFLD has become a critical problem for public health, because of the involvement of other collateral cardiometabolic diseases, including diabetes and hypertension [8,9,10,11,12]. NAFLD patients are at a risk of progressing to non-alcoholic steatohepatitis (NASH) and ultimately cirrhosis; they are also at a higher risk of cardiovascular diseases (CVD), including coronary heart disease and stroke [13]. NAFLD confers increased cardiovascular disease risk independent of traditional cardiovascular risk factors and metabolic syndrome [14]. The abnormal accumulation of lipids in the liver causes non-alcoholic steatohepatitis (NASH) with progressive liver damage characterized by inflammation and oxidative stress, that could lead to advanced fibrosis or cirrhosis [15,16,17]. In the case of repeated damage, liver parenchyma could respond with an excessive extracellular matrix (ECM) accumulation, due to activation of hepatic stellate cells (HSCs) in perisinusoidal space [18,19]. After differentiation in myofibroblast-like cells, HSCs play a key role in ECM remodeling through the overexpression of α-smooth muscle actin (α-SMA), that leads to hepatic fibrosis [20,21,22,23]. An imbalance of ECM synthesis and degradation is caused by the activity of many mediators, such as mitogen-activated protein kinase (MAPK), integrins and various growth factors [24,25]. Transforming growth factor β (TGF-β) is a key mediator in fibrotic matrix increase [26] as the main pro-fibrogenic cytokine, promotes the accumulation of ECM through both activation of SMAD-dependent and independent pathways, and regulation of enzymes like metalloproteinases [27,28,29,30]. The progression of NAFLD and the development of hepatic fibrosis is caused by changes in redox balance, leading to an increase of reactive oxygen species (ROS) levels [31,32,33]. Heme oxygenase (HO) is a microsomal enzyme involved in oxidative stress control, that catabolizes heme into biliverdin, ferrous iron (Fe^2+^) and carbon monoxide (CO) [34,35,36]. HO exerts an antioxidant effect through its products, which possess many biological protective properties involved in the regulation of inflammation and apoptosis [37]. Cytoprotective actions of HO and its by-products can be harmful, especially when translated into pathophysiological processes like tumorigenesis [38,39,40,41,42]. 

Humans own two isoenzymes of HO, namely HO-1 and HO-2, encoded by the HMOX1 and HMOX2 genes, respectively. Also known as heat shock protein 32, HO-1 is induced in a range of cells and in several organs, in response to inflammation and oxidative stress, while HO-2 is constitutively expressed [43,44,45,46,47]. It has been shown that HO-1 ensures a protective effect on liver cells under injury conditions, and an induction of HO-1 is even involved in the prevention of liver fibrosis development [48,49]. Conversely, low levels of HO-1 are related to severe oxidative stress and organ failure, showed by iron deposits in the damaged liver [50]. The aim of this study is to investigate the role of HO in two of the main processes involved in the NASH pathology, using the human hepatocellular carcinoma cell line (HepG2) treated with free fatty acids (FFA), and the human hepatic stellate cells (LX2) treated with TGF-β, as steatosis and fibrosis models, respectively.

## 2. Materials and Methods

### 2.1. Cell Culture

HepG2 cells retain many characteristics of normal differentiated quiescent hepatocytes. They were widely used in several studies as NAFLD in vitro model, administrating fatty acids [51,52,53]. HepG2 cells were maintained in DMEM supplement with 10 % FBS, 1 % Penicillin and Streptomycin solution and incubated at 37 °C in a 5 % CO_2_ humidified atmosphere. For the experiments, the cells were seeded in 24-well plates at a density of 5 × 10^5^ cells per well. Then the cells were treated for 24 h with DMEM containing FFA 2 mM (palmitic acid and oleic acid 2:1) in the presence or absence of Tin- Mesoporphyrin IX (SnMP) 5 μM, alone or in combination with Cobalt Protoporphyrin (CoPP) 5 μM, for 2 h as pre-treatment.

The LX2 cells were maintained in medium DMEM low glucose supplemented with 10% FBS and 1% Penicillin and Streptomycin solution. For this study, the cells were seeded in 6-well plates and then treated with 5 ng/mL TGF-β to induce their activation and the collagen release, in the presence or absence of SnMP 5 μM and CoPP 5 μM.

### 2.2. Oil Red O Staining

Staining was performed using 0.21% Oil Red O in 100% isopropanol (Sigma-Aldrich, St. Louis, MO, USA). Briefly, hepatocytes were fixed in 10% formaldehyde, stained with Oil Red O for 10 minutes, rinsed with 60% isopropanol (Sigma-Aldrich), and the Oil Red O eluted by adding 100% isopropanol for 10 mins and the optical density (OD) measured at 490 nm, for 0.5 sec reading. Lipid droplets accumulation was examined by using an inverted multichannel LED fluorescence microscope (Evos, Life Technologies, Grand Island, NY, USA).

### 2.3. ROS Measurement

Determination of ROS was performed by using a fluorescent probe 2′,7′-dichlorofluorescein diacetate (DCFH-DA); 100 μM DCFH-DA, dissolved in 100% methanol which was added to the cellular medium and the cells were incubated at 37 °C for 30 min. Under these conditions, the acetate group is not hydrolyzed [54]. The fluorescence [corresponding to the oxidized radical species 2′,7′-dichlorofluorescein (DCF)] was monitored spectrofluorometrically (excitation, λ = 488 nm; emission, λ = 525 nm). The total protein content was evaluated for each sample, and the results were reported as percentage increase in fluorescence intensity (FI)/mg protein with respect to control untreated cells.

The quantitative measurement of cellular populations undergoing oxidative stress was performed using the Muse Oxidative Stress Kit (Merck Millipore, Billerica, MA, USA), according to the manufacturer’s instructions. This assay utilizes dihydroethidium (DHE), which is cell membrane-permeable and, upon reaction with superoxide anions, undergoes oxidation to form DNA-binding fluorophore. The kit determines the percentage of cells that are negative [ROS(−)] and positive [ROS(+)] for reactive oxygen species. The count and percentage of cells undergoing oxidative stress were quantified using the Muse Cell Analyzer and Muse analysis software (Merck Millipore, Milano, Italy). 

### 2.4. Sircol Collagen Assay

Total soluble collagen in cell culture supernatants was quantified using the Sircol collagen assay (Biocolor, Belfast, UK). For these experiments, confluent cells in 6-plate wells were incubated for 24 h with 5 ng/mL of TGF-β (Sigma). One mL of Sirius red stain, an anionic dye that reacts specifically with basic collagen side chain groups, was added to 400 μL of supernatant and incubated with gentle rotation for 30 min at room temperature. After centrifugation at 12,000 g for 10 min, the collagen-bound dye was dissolved again after the addition of 1 mL of 0.5 M NaOH and absorbance at 540 nm was measured using a microplate spectrophotometer reader (Synergy HT, BioTek). The absorbance was directly proportional to the amount of newly formed collagen in the cell culture supernatant.

### 2.5. RNA Extraction and qRT-PCR

RNA was extracted by Trizol reagent (Invitrogen, Carlsbad, CA, USA) [4]. First strand cDNA was then synthesized with Applied Biosystem (Foster City, CA, USA) reverse transcription reagent. Quantitative real-time PCR was performed in Step One Fast Real-Time PCR System Applied Biosystems using the SYBR Green PCR MasterMix (Life Technologies) [5]. The specific PCR products were detected by the fluorescence of SYBR Green, the double stranded DNA binding dye. The relative mRNA expression level was calculated by the threshold cycle (Ct) value of each PCR product and normalized with that of GAPDH by using comparative 2^–ΔΔCt^ method.

### 2.6. Statistical Analyses

Statistical significance (*P* < 0.05) of differences between experimental groups was determined by the Fisher method for analysis of multiple comparisons. For comparison between treatment groups, the null hypothesis was tested by either the single-factor analysis of variance (ANOVA) for multiple groups, or the unpaired t-test for two groups, and the data are presented as mean ± SD.

## 3. Results

### 3.1. Effect of HO Inhibition on Hepatic Fatty Storage

HepG2 cells were treated with FFA 2 mM in order to create an in vitro model of hepatic steatosis.

After 24 h an Oil Red O staining was performed to evaluate the amount of lipid droplets. Figure 1 shows the FFA treatment was able to increase the triglycerides storage in HepG2 cells compared to the control untreated group. The group pre-treated with SnMP showed a significative increase of the lipid droplets amount compared to the CTRL and FFA groups, while in the group pre-treated with the combination of SnMP and CoPP this effect was partially reversed, indicating the involvement of the HO system in the lipid metabolism regulation.

### 3.2. Effect of HO Regulation on Lipid Metabolism Pathway

To support the Oil Red data, we analyzed the mRNA levels of lipid metabolism pathway genes. Figure 2A–C showed an increase of Diglyceride acyltransferase 1 (DGAT-1), Sterol regulatory element-binding transcription factor 1 (SREBP-1) and Fatty acid synthetase (FAS) gene expression in the group with SnMP treatment compared to the control group, suggesting an increased synthesis of cholesterol, fatty acids and triglycerides. The co-treatment with CoPP, a strong inducer of HO-1 expression, reversed the SnMP effect on DGAT-1 and SREBP-1 genes, but did not affect FAS expression. We also analyzed the HO-1 levels (Figure 2D), that were increased in the FFA group when compared with the control group, probably because the FFA treatment causes a moderate oxidative stress. As expected, in the SnMP group, HO-1 was markedly increased compared to the FFA group because, as previous studies demonstrated, SnMP decreased the HO activity, but increased its protein expression [55,56]. The group with both compound SnMP and CoPP showed a synergic effect with a strong increase of HO-1 expression compared with all other groups. Sirtuin 1 expression (SIRT1) did not show any difference in mRNA expression, in both groups FFA and FFA-SnMP when compared with the control, but the co-administration of CoPP showed a significant increase compared with other groups (Figure 2E). That result is in accordance with several published studies that demonstrated the positive relation between the HO and SIRT1 genes expression [57].

### 3.3. Hepatic Fibrosis In Vitro Model

We investigated another main process that characterized the NAFLD physiopathology creating an in vitro model of hepatic fibrosis, administrating to LX2 cells the TGF-β protein, that is known to activate fibroblasts resulting in collagen release [58]. We treated LX2 cells with 5 ng/mL of TGF-β and we measured the reactive oxygen species (ROS) generation and the soluble collagen release. ROS production was increased by 72% after 1 h in the group with TGF-β 5 ng compared to the control group (Figure 3A). We measured the soluble collagen using a colorimetric kit (Sircol) and we showed a significant increase of collagen release in the group treated with TGF-β 5 ng compared to the control group. Furthermore, we analyzed the mRNA expression of the main genes involved in the collagen production as collagen type 1 alpha 1 (COL1A1), alpha smooth muscle actin (α-SMA), SMAD3, SMAD4, SMAD7 and TIMP-1. As shown in Figure 4, in all the genes the expression was markedly increased by TGF-β 5ng treatment compared to the control group. 

### 3.4. Effect of HO Inhibition on ROS Generation and Soluble Collagen Release

We investigated the role of HO in this model of fibrosis using SnMP and CoPP, both at the concentration of 5 μm. In order to confirm the role of ROS in liver fibrosis and if ROS production is regulated by treatment of SnMP with or without CoPP, we evaluated ROS production in LX2 cells by flow-cytometer (Figure 5A). The obtained results showed that SnMP treatment significantly increased ROS generation but co-treatment with CoPP reversed the effect mediated by SnMP (Figure 5B). To investigate the correlation between ROS and fibrosis, we treated LX2 cells with SnMP and CoPP for 2 h before the administration of TGF-β 5ng/ml and we obtained a significant increase in collagen release levels after 3 h in the group with SnMP compared to the TGF-β group (Figure 5C). The effect was reversed by the HO-1 inducer CoPP.

## 4. Discussion

Considering the complexity of NAFLD and its rising prevalence globally, it is of primary importance to find new protein targets for the regulation of the pathways involved in this pathology. In NAFLD, an increase in hepatic FFAs uptake, lipid synthesis, impaired β-oxidation, and a decrease in lipid export facilitates accumulation of fat in the liver [59,60]. In order to study the role of the HO system in the main NASH pathological aspects, we propose two different in vitro models for steatosis and fibrosis. Primary hepatocytes, derived from human liver samples, are an ideal in vitro model for studying hepatosteatosis, but the difficulty to obtain normal clinical liver samples lead us to use HepG2 cells as an alternative cellular model [52]. Whereas oleic and palmitic acid represent the main fatty acids in the triglycerides (TG) content of steatotic patients, we treated HepG2 cells with a combination of these fatty acids to simulate NAFLD [61]. To assess how HO affects the hepatocytes lipid metabolism, we cultured the HepG2 cells with a well-known HO activity inhibitor named SnMP, alone or in combination with the strong HO inducer CoPP. Figure 1 shows that FFA was able to increase the intracellular lipid droplets content compared to untreated cells, in particular in the cells that received the pre-treatment with SnMP, suggesting that HO inhibition impairs the lipid metabolism in hepatocytes. SREBP, characterized by the three isoforms SREBP-1a, SREBP-1c and SREBP-2, plays a key role on the regulation of various genes expression involved in cholesterol and lipid metabolism. SREBP-1c represents the major isoform that controls FA synthesis in the liver and is regulated by a series of nutritional and hormonal stimulus through transcriptional and post-transcriptional mechanisms. Yahagi et al. showed that knockout SREBP-1c ob/ob mice presented a significant reduction in the hepatic expression of lipogenic genes preventing liver steatosis [62]. Conversely, overexpression of SREBP-1c results in raised levels of FAS, acetyl CoA carboxylase (ACC) and stearoyl-CoA desaturase (SCD) causing an increase in lipogenesis [63] that, in concert with an augmented hepatic FFAs uptake, is known to contribute TG accumulation in the hepatocytes [64]. In HepG2 cells treated with FFA, we observed a significant increase of SREBP-1c, DGAT-1 and FAS levels after SnMP treatment which is associated with an increase of fatty acid storage. Conversely, co-administration of CoPP and SnMP reversed DGAT and SREBP-1c mRNA levels, confirming that HO can affect TG formation and storage in hepatocyte’s cytoplasm. As the most extensively studied sirtuin, Sirt1 has a prominent role in metabolic tissues, such as the liver, skeletal muscle and adipose tissues. Sirt1 overexpression in the liver can deacetylate a range of substrates, including SREBP-1c, PGC-1α and FoxO1 proteins, and can result in a pronounced effect on glucose and lipid metabolism [65,66]. Previous studies indicated that the overexpression of Sirt1 protects against HFD-induced hepatic inflammation by decreasing the NFκB-mediated induction of inflammatory cytokines [67]. Consistent with previous published results, our data showed a positive regulation of Sirt1 by HO-1 induction [68]. Despite SnMP not affecting Sirt1 gene levels, induction of HO-1 by CoPP treatment significantly increased Sirt1 mRNA. The crosstalk between HO-1 and Sirt1 may be considered as a pivotal axis against oxidative stress caused by hyperglycemia and hyperlipidemia, and it is essential to protect the liver from steatosis. The results reported here extend our previous findings that upregulation of HO-1 in hepatocytes results in the negative regulation of lipogenesis [68,69,70,71]. In order to evaluate the detrimental role of HO-1 inhibition on liver fibrosis, we first established an in vitro model of liver fibrosis using TGF-β as a fibrotic agent in human stellate hepatic cells. In chronic liver diseases, hepatic stellate cells have been considered as a primary target for active TGF-β, thereby cell treatment with TGF-β contributes to their activation and subsequent fibrogenesis. The increase of ROS levels mediated by TGF-β treatment (Figure 3) and lipid peroxidation products contribute to collagen release causing the onset and progression of fibrosis [72]. Blockage of ROS generation by HSC in response to TGF-β and alleviation of the downstream proteins is a strategy to inhibit liver fibrosis [73].

As shown in Figure 4, the induction of α-SMA, COL1A1 and SMADs represents reliable markers of HSCs activation to myofibroblast-like cells with direct contribution to hepatic fibrogenesis. Heme oxygenase, known to be a powerful antioxidant enzymatic system, plays a key role in redox balance by counteracting ROS production [74]. Liu et al. showed that CoPP upregulates HO-1 and other oxidative stress-responsive genes expression and decreases mitochondria-derived ROS production [75]. Contrarily, SnMP increases ROS and oxidative stress [76]. Consistent with these findings, in the present article, we showed that HO activity inhibition further increased ROS and collagen release (Figure 5) from activated LX2 compared to cells treated exclusively with TGF-β. Furthermore, we found that increased HO-1 levels by CoPP reversed the effect mediated by SnMP and reduced the levels of ROS and soluble collagen released from activated LX2. These novel findings underscore the importance of targeting HO-1 and provide additional evidences for a link between liver disorders and the HO system. A limitation of the in vitro study, but in agreement with the report by Schulz et al. [76], although SnMP treatment decreased cellular HO activity, the combination of SnMP+CoPP did not increase the enzymatic activity compared to the SnMP group (data not shown). However, it must be taken into consideration that induction of HO-1 expression presents several non-canonical functions not associated to the enzymatic activity [77]. Consequently, the increase of HO-1 levels may provide a therapeutic approach to address the metabolic alterations associated with NAFLD and its progression to NASH.

## Figures and Tables

**Figure 1 antioxidants-08-00277-f001:**
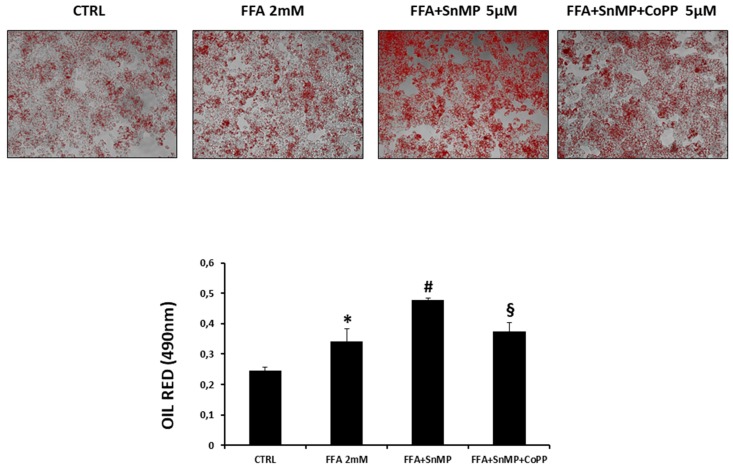
Effect of HO inhibition on oil droplets formation in hepatocytes. We measured the effect of SnMP 5 μM treatment on lipogenesis in the presence of FFA. **p* < 0.05 vs. CTRL, # *p* < 0.05 vs. FFA 2mM, § *p* < 0.05 vs. FFA+SnMP.

**Figure 2 antioxidants-08-00277-f002:**
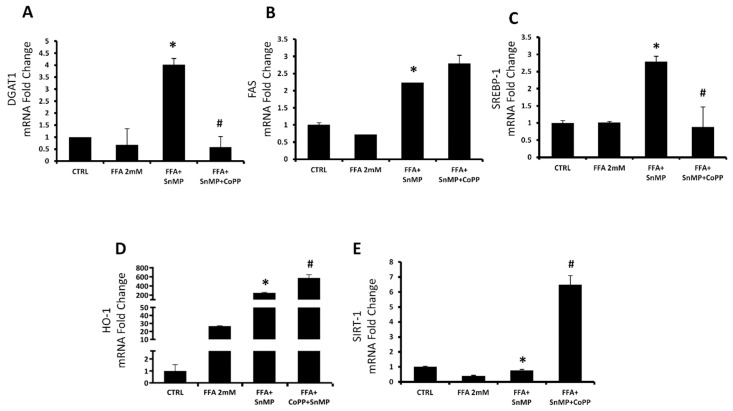
mRNA expression of DGAT1 (**A**) FAS, (**B**) SREBP-1, (**C**) HO-1, and (**D**) SIRT1, (**E**) of HepG2 control cells, cells treated with FFA 2 mM and cells treated with SnMP 5 uM alone or in combination with CoPP 5 uM. Results are mean ± SD, * *p* < 0.05 vs. FFA 2mM, ^#^
*p* < 0.05 vs. FFA+SnMP.

**Figure 3 antioxidants-08-00277-f003:**
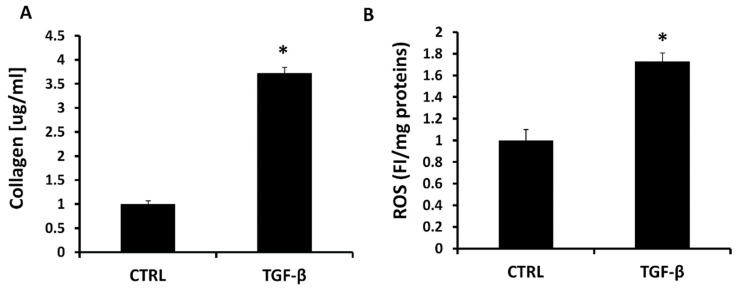
(**A**) Soluble collagen in LX2 cells activated with TGF-β 5 ng for 24 h. Soluble collagen measurement are expressed as μg/mL. Values represent the means ± SD of three experiments performed in triplicate. * *p* < 0.05, significant result vs. untreated LX2 cells. (**B**) Intracellular oxidants in LX2 cells activated with TGF-β 5ng for 1 h. Results are mean ± SD, * *p* < 0.05 vs. CTRL.

**Figure 4 antioxidants-08-00277-f004:**
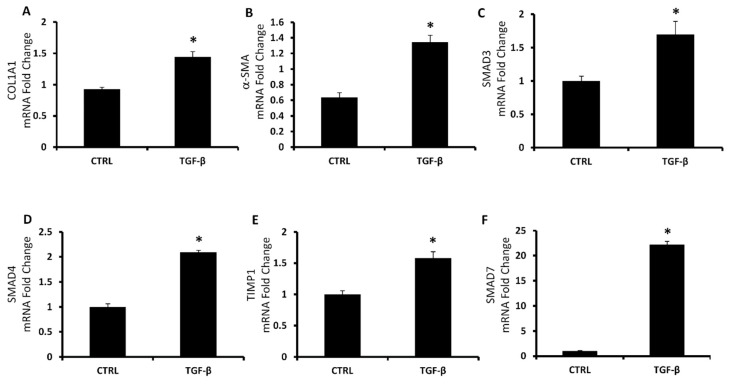
mRNA expression of fibrosis pathway. COL1A1 (**A**), α-SMA (**B**), SMAD3 (**C**), SMAD4 (**D**), TIMP1 (**E**) and SMAD7 (**F**) of LX2 control cells and cells treated with TGF-β 5 ng. Results are mean ± SD, * *p* < 0.05 vs. CTRL.

**Figure 5 antioxidants-08-00277-f005:**
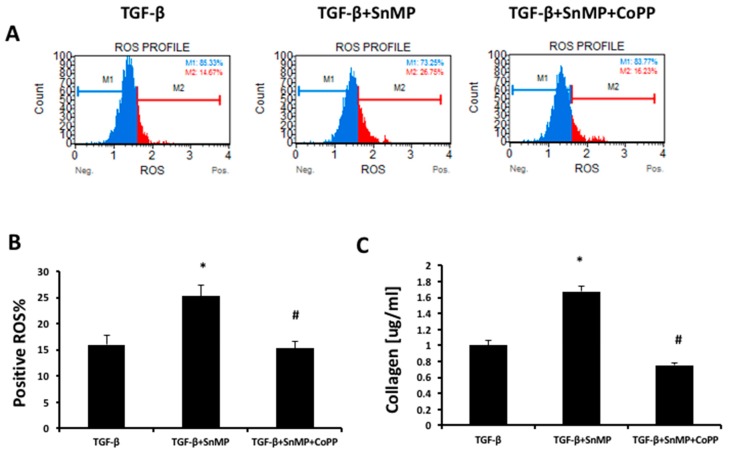
(**A**) The quantitative measurement of cells undergoing oxidative stress was evaluated by cytometry, using the Muse Oxidative Stress Kit. Cells were pre-treated with SnMP (5 μM) and CoPP (5 μM) for 24 h and then incubated for 1h with TGF-β 5 ng/mL. (**B**) The graph showed the positive ROS percentage in the different groups. * *p* < 0.05 vs. TGF-β, ^#^
*p* < 0.05 vs. TGF-β+SnMP. (**C**) Soluble collagen in LX2 cells treated with TGF-β 5 ng/mL for 24 h in the presence or absence of SnMP 5 μM alone or in combination with CoPP 5 μM. Soluble collagen measurement is expressed as μg/mL. Values represent the means ± SD of three experiments performed in triplicate. * *p* < 0.05 vs. TGF-β; # *p* < 0.05 vs. TGF-β+SnMP.

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
