# Peer review of "Inhibition of Heme Oxygenase Antioxidant Activity Exacerbates Hepatic Steatosis and Fibrosis In Vitro"

_antioxidants, 2019, doi:10.3390/antiox8080277_

Round 1
Reviewer 1 Report
Authors clearly show HO regulates lipid storage and collagen production in hepatocytes, however, I have some concern about m
ethods and results.
1.In Fig1, did you look into the effect of CoPP on oil droplets formation in hepatocytes? Please show the data.
2.In Fig2D, authors show HO-1mRNA is increased by combination of SnMP and CoPP. However, it does not mean incresed HO activi
ty. Please add the data to show enhanced HO activity by combined treatment of SnMP and CoPP.
3.Fig3B shows that ROS production is enhanced by TGF-b in LX2 cells, and TGF-b treatment regulates mRNA expression of fibros
is pathway shown in Fig4. Does increased ROS mediate effect of TGF-b on each fibrosis pathway? Does scavenger of ROS cancel
led effect of TGF-b on fibrosis pathway? Are these fibrosis pathway affected equally by ROS or is there any specific pathway
strongly affeced by ROS?
4.Fig5 shows soluble collagen contents in LX2 cells is affected by SnMP in the absence or presence of CoPP. Does ROS regula
te collagen contents? Authors should examine if ROS production is regulated by treatment of SnMP with or without CoPP.
5.Related to above question, does SIRT-1 mediate soluble collagen contents in LX2 cells? Authos should show the data indicat
ing that inhibition of SIRT-1 cancelled effects of SnMP and CoPP on soluble collagen in cells.
Author Response
We thank the reviewer for the precious comments which significantly improved the revised manuscript. The changes in the text are highlighted in yellow.
Reviewer 1
Authors clearly show HO regulates lipid storage and collagen production in hepatocytes, however, I have some concern about methods and results.
1.In Fig1, did you look into the effect of CoPP on oil droplets formation in hepatocytes? Please show the data.
The requested data has been added to the manuscript.
2.In Fig2D, authors show HO-1mRNA is increased by combination of SnMP and CoPP. However, it does not mean incresed HO activity. Please add the data to show enhanced HO activity by combined treatment of SnMP and CoPP.
This is a good point. As suggested by the reviewer we performed HO activity for the different groups. Although SnMP treatment decreased cellular HO activity, the combination SnMP+CoPP did not reverse the enzymatic activity compared to SnMP group (data not shown). These results are in agreement with previous studies reported by Schulz et al. (“Metalloporphyrins - an update” Schulz et al. Front Pharmacol. 2012 Apr 26;3:68). Protoporphyrins with cobalt, iron, or cadmium as central metals have been found to induce HO; but only iron containing metalloporphyrins, such as heme (FePP), act as actual substrates. CoPP is a unique metalloporphyrin exhibiting a dualism: significantly inhibiting HO activity in vitro and enhancing HO activity in vivo due to its strong activation of HO-1 gene expression. CoPP might be considered as a potential therapeutic agent since it increases Nrf2 levels and increases the degradation of Bach1, a negative transcriptional factor of Heme Oxygenase (“Role of Bach1 and Nrf2 in up-regulation of the heme oxygenase-1 gene by cobalt protoporphyrin” Shan et al. FASEB J. 2006 Dec;20(14):2651-3). Additionally, it has been shown that HO presents several non-canonical functions not associated to the enzymatic activity (“The non-canonical functions of the heme oxygenases” Vanella et al. Oncotarget. 2016 Oct 18;7(42):69075-69086).
3.Fig3B shows that ROS production is enhanced by TGF-b in LX2 cells, and TGF-b treatment regulates mRNA expression of fibrosis pathway shown in Fig4. Does increased ROS mediate effect of TGF-b on each fibrosis pathway? Does scavenger of ROS cancelled effect of TGF-b on fibrosis pathway? Are these fibrosis pathway affected equally by ROS or is there any specific pathway strongly affected by ROS?
Transforming growth factor β (TGF-β) is considered to be the most potent profibrogenic cytokine. TGF-β stimulates the production of reactive oxygen species (ROS) in various types of cells. It has been reported that TGF-β stimulated ROS production through activation of cell-membrane associated oxidase, which led to an increased release of H2O2 to the extracellular space. Another study also showed that TGF-β increased ROS production in mitochondria in rat hepatocytes (“Oxidative stress and glutathione in TGF-β-mediated fibrogenesis.” R.-M.LiuK.A.Gaston Pravia. Free Radical Biology and Medicine Volume 48, Issue 1, 1 January 2010, Pages 1-15). Cucoranu et al. showed that ROS mediated TGF-β-induced fibrosis by activating Smad2/3, although the detailed mechanism underlying Smad2/3 activation by ROS was not explored (“NAD(P)H Oxidase 4 Mediates Transforming Growth Factor-β1–Induced Differentiation of Cardiac Fibroblasts Into Myofibroblasts” Cucoranu et al. Circulation Research. 2005;97:900–907). Moreover, blockage of ROS generation by HSC in response to TGF-β and alleviation of the downstream proteins is a strategy to inhibit liver fibrosis (“Prevention of TGF-β-induced early liver fibrosis by a maleic acid derivative anti-oxidant through suppression of ROS, inflammation and hepatic stellate cells activation”Yang et al. PLoS One. 2017 Apr 6;12(4):e0174008).
4.Fig5 shows soluble collagen contents in LX2 cells is affected by SnMP in the absence or presence of CoPP. Does ROS regulate collagen contents? Authors should examine if ROS production is regulated by treatment of SnMP with or without CoPP.
Garcia-Trevijano and Cao showed that TGF-β1 increased ROS production and induced procollagen-α1(I) mRNA expression in hepatic stellate cells, whereas treatment of cells with catalase reduced ROS levels and prevented the induction of procollagen by TGF-β, suggesting that ROS mediate the TGF-β induction of procollagen expression in these cells (“Transforming growth factor beta1 induces the expression of alpha1(I) procollagen mRNA by a hydrogen peroxide-C/EBPbeta-dependent mechanism in rat hepatic stellate cells” García-Trevijano et al., Hepatology. 1999 Mar; 29(3):960-70. “DLPC decreases TGF-beta1-induced collagen mRNA by inhibiting p38 MAPK in hepatic stellate cells.” Cao Q, Mak KM, Lieber CS Am J Physiol Gastrointest Liver Physiol. 2002 Nov; 283(5):G1051-61).
SnMP and CoPP are respectively a strong inhibitor and inducer of the enzyme Heme Oxygenase, known to be a powerful antioxidant enzymatic system that plays a key role in redox balance by counteracting ROS production (“Redox Functions of Heme Oxygenase-1 and Biliverdin Reductase in Diabetes” Rochette et al. Trends Endocrinol Metab. 2018 Feb;29(2):74-85). Liu et al., showed that CoPP upregulates HO-1 and other oxidative stress-responsive genes expression and decreases mitochondria-derived ROS production (“Cobalt Protoporphyrin Induces HO-1 Expression Mediated Partially by FOXO1 and Reduces Mitochondria-Derived Reactive Oxygen Species Production” Liu et al., PLoS One. 2013; 8(11): e80521). Contrarily, SnMP increases ROS and oxidative stress (“The Role of Heme Oxygenase 1 in the Protective Effect of Caloric Restriction against Diabetic Cardiomyopathy” Waldman et al. Int. J. Mol. Sci. 2019, 20(10), 2427).
5.Related to above question, does SIRT-1 mediate soluble collagen contents in LX2 cells? Authors should show the data indicating that inhibition of SIRT-1 cancelled effects of SnMP and CoPP on soluble collagen in cells.
A recent study showed that stimulation of LX-2 cells with TGF-β1 resulted in a significant suppression of SIRT1 protein. Nevertheless, TGF-β1-induced LX-2 cell activation was inhibited by SIRT1, and this was accompanied by up-regulation of cell apoptosis-related proteins. Overexpression of SIRT1 also attenuated TGF-β1-induced expression of myofibroblast markers α-SMA and collagen type I alpha 1 ameliorating liver fibrosis. (“Silent information regulator 1 (SIRT1) ameliorates liver fibrosis via promoting activated stellate cell apoptosis and reversion” Wu et al., Toxicol Appl Pharmacol. 2015 Dec 1;289(2):163-76). Activation of SIRT1 decreases fatty liver by a reducing expression of lipogenic enzymes (“Treatment with SRT1720, a SIRT1 activator, ameliorates fatty liver with reduced expression of lipogenic enzymes in MSG mice” Yamazaki et al., Am J Physiol Endocrinol Metab 2009;297:E1179–E1186). Futhermore, recent studies reported that HO-1 induction attenuates hepatic lipid deposition, prevents the development of hepatic fibrosis and abates NAFLD-associated vascular dysfunction. Those effects are mediated by activation of SIRT1 gene expression, suggesting an axis between HO-1 and SIRT1. (“Fructose Mediated Non-Alcoholic Fatty Liver Is Attenuated by HO-1-SIRT1 Module in Murine Hepatocytes and Mice Fed a High Fructose Diet” Sodhi et al. PLoS One. 2015 Jun 22;10(6):e0128648).
We investigated the SIRT1 levels in relation with its ability to mediate Heme Oxygenase action on lipid metabolism. The aim of this short communication is to show that down-regulation of Heme Oxygenase activity affect lipid metabolism and collagen release in steatosis and fibrosis in vitro model.
Reviewer 2 Report
The authors present an interesting study modeling heme oxygenase inhibition, using Tin 20 Mesoporphyrin IX (SnMP), with direct effect over the antioxidant activity, lipid metabolism and fibrosis process, resulting in an increased lipid storage and collagen release in hepatocytes. They used ROS measurement, RT-PCR and soluble collagen assay to assess the intracellular oxidant, the fibrosis pathway and the soluble collagen in LX2 cells. The article it is an original contribution which has many good features including the experimental design and accuracy.
Author Response
The authors present an interesting study modeling heme oxygenase inhibition, using Tin 20 Mesoporphyrin IX (SnMP), with direct effect over the antioxidant activity, lipid metabolism and fibrosis process, resulting in an increased lipid storage and collagen release in hepatocytes. They used ROS measurement, RT-PCR and soluble collagen assay to assess the intracellular oxidant, the fibrosis pathway and the soluble collagen in LX2 cells. The article it is an original contribution which has many good features including the experimental design and accuracy.
We thank you for your positive comment about our manuscript.
Reviewer 3 Report
HepG2 and LX2 cells were used as NAFLD model. The disease is a multifactorial one, as evidenced also by a series o omics studies (Curr Opin Clin Nutr Metab Care. 2019 Jun 18; Metabolites. 2019 Apr 11;9(4). pii: E70.; OMICS. 2019 Apr;23(4):181-189; Metabolites. 2019 Feb 1;9(2). pii: E25.; Cell Syst. 2018 Jan 24;6(1):7-9; J Lipid Res. 2015 Mar;56(3):722-36. Sci Data. 2015 Dec 8;2:150068)
Here the authors focus on a specific aspect. Relations are established between HO-1 activity/inhibition or treatment with TGF-β 5ng, on one side, and oil droplets formation, collagen release, mRNA expression of lipid metabolism pathway genes, on the other. The bar plots of Figures 1-5 show significant changes.
The text requires some language revision, as exemplified below.
Line 33: “health, because of the involving of other collateral cardiometabolic diseases, including …”
Line 40: “In case of repeated damage, liver parenchyma could responde …”
Line 52: “ that catabolizes heme into biliverdin, ferrous iron (Fe2+) and carbon monoxide (CO)”
Lines 54-55: In the sentence ”Cytoprotective actions of HO-1 and its by-products can be harmful, …” HO-1 is not yet defined. Do the authors really mean that HO-1 can be harmful but HO-2 no?
Lines 151-152: the sentence is unclear and should be rephrased “and we showed that following SnMP treatment the all genes level were overexpressed compared to control group”
Line 155: “We also analyzed HO-1 levels (Figure 2D-E), that were increased in …”
Line 206: “find new protein targets ...”
Graphics
Fig. 2: I imagine that a fold change of 1 has to be assumed for Ctr even when not visible with the selected vertical scale
Author Response
We thank the reviewer for the precious comments which significantly improved the revised manuscript. The changes in the text are highlighted in yellow.
HepG2 and LX2 cells were used as NAFLD model. The disease is a multifactorial one, as evidenced also by a series o omics studies (Curr Opin Clin Nutr Metab Care. 2019 Jun 18; Metabolites. 2019 Apr 11;9(4). pii: E70.; OMICS. 2019 Apr;23(4):181-189; Metabolites. 2019 Feb 1;9(2). pii: E25.; Cell Syst. 2018 Jan 24;6(1):7-9; J Lipid Res. 2015 Mar;56(3):722-36. Sci Data. 2015 Dec 8;2:150068)
Thank you for your suggested references. They have been added.
Here the authors focus on a specific aspect. Relations are established between HO-1 activity/inhibition or treatment with TGF-β 5ng, on one side, and oil droplets formation, collagen release, mRNA expression of lipid metabolism pathway genes, on the other. The bar plots of Figures 1-5 show significant changes.
The text requires some language revision, as exemplified below.
Line 33: “health, because of the involving of other collateral cardiometabolic diseases, including …”
It has been fixed.
Line 40: “In case of repeated damage, liver parenchyma could responde …”
It has been fixed.
Line 52: “ that catabolizes heme into biliverdin, ferrous iron (Fe2+) and carbon monoxide (CO)”
It has been fixed.
Lines 54-55: In the sentence ”Cytoprotective actions of HO-1 and its by-products can be harmful, …” HO-1 is not yet defined. Do the authors really mean that HO-1 can be harmful but HO-2 no?
The sentence has been changed as follow: “Cytoprotective actions of HO and its by-products can be harmful,..”
Lines 151-152: the sentence is unclear and should be rephrased “and we showed that following SnMP treatment the all genes level were overexpressed compared to control group”
The sentence has been changed as follow: “To support the Oil Red data, We analyzed the mRNA levels of lipid metabolism pathway genes. Figures 2A-C showed an increase of Diglyceride acyltransferase 1 (DGAT-1), Sterol regulatory element-binding transcription factor 1 (SREBP-1) and Fatty acid synthetase (FAS) gene expression in the group with SnMP treatment compared to control group, suggesting an increased synthesis of cholesterol, fatty acids and triglycerides.”
Line 155: “We also analyzed HO-1 levels (Figure 2D-E), that were increased in …”
It has been fixed.
Line 206: “find new protein targets ...”
It has been fixed.
Graphics
Fig. 2: I imagine that a fold change of 1 has to be assumed for Ctr even when not visible with the selected vertical scale
The graph vertical scale of Fig.2-D has been changed to show the CTRL bar.
Round 2
Reviewer 1 Report
Authors responded to reviewer's comments and suggestions, and added new result in Fig1. However, most part of manuscript, including main text, figures and references are not changed.
I have requested authors to perform additional experiments followed by comments in this setting to improve the manuscript, but not just in literature .
Author Response
Authors responded to reviewer's comments and suggestions, and added new result in Fig1. However, most part of manuscript, including main text, figures and references are not changed.
I have requested authors to perform additional experiments followed by comments in this setting to improve the manuscript, but not just in literature.
We thank the reviewer for his comments and efforts in improving the scientific quality of our manuscript. We highlighted in green the new changes in the text.
In order to confirm the role of ROS in liver fibrosis and if ROS production is regulated by treatment of SnMP with or without CoPP, as suggested by the reviewer, we evaluated ROS production in LX2 cells by flow-cytometer (new Figure 5). The obtained results showed that SnMP treatment significantly increased ROS generation but co-treatment with CoPP reversed the effect mediated by SnMP. These results correlate with collagen production suggesting that inhibition of heme oxygenase activity further increase TGF-b mediated-ROS levels which induce a significant increase in collagen release.
Scavenging ROS by cell treatment with CoPP alone (Positive ROS 7.6 %), blocked the effect of TGF-b (Positive ROS 15.9%) on ROS production (data not shown).
Round 3
Reviewer 1 Report
Authors added new data to support their hypothesis and scientific quality of the revised manuscript is much improved.